# Development of FT-NIR Technique to Determine the Ripeness of Sweet Cherries and Sour Cherries

**Marietta Fodor** 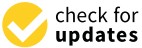

Department of Food and Analytical Chemistry, Institute of Food Science and Technology, Hungarian University of Agriculture and Life Sciences, Villányi út 29-43, 1118 Budapest, Hungary; fodor.marietta@uni-mate.hu

**Abstract:** The FT-NIR technique was used for the rapid and non-destructive determination of sweet cherry and sour cherry ripeness. Titratable acidity (A), water-soluble total solids (SSC), total anthocyanin (TA) content and calculated maturity index (SSC/A = MI) were used as reference values. PLS correlations were validated by seven-fold cross-validation (RMSECV for different parameters: DM = 1.25%, $w/w$; A = 0.14%, $w/w$; SSC = 0.97%, $w/w$; TA = 17.5 g/100 mL; MI = 1.66) and test-validation (RMSEP for different parameters: DM = 1.46%, $w/w$; A = 0.19%, $w/w$; SSC = 0.99%, $w/w$; TA = 17.5 g/100 mL; MI = 1.59). Different discriminant analyses and support vector machine (SVM) classifications were performed for each parameter. The quadratic discriminant analysis (QDA) was found to be the best pattern recognition method. A maturity degree (MD) was developed based on the reference values, which classified the samples into mature and immature categories with an accuracy of 98.44%.

**Keywords:** FT-NIR; PLS; LDA; QDA; SVM; maturity degree

## 1. Introduction

Spectral imaging (SI) and non-destructive techniques (hyperspectral imaging—HSI; near-infrared spectroscopy—NIR; electronic nose EN; electronic tongue—ET) are becoming more and more common in analytical studies due to the development of technology and chemometric data processing. This is particularly true in the food industry.

The primary problem in the food industry is the complex matrix and the matrix dependence of the applicability of the methods due to their nature.

This is particularly true for the analysis of a concept such as fruit ripeness. External characteristics such as color are often misleading information carriers [1].

Determining the stage of ripeness of fruit is a key factor in determining its harvest time, storability and ripening quality. To ensure flexibility in marketing and to guarantee an acceptable eating quality for the customer, it is very important to determine the correct stage of ripeness. Maturity indicators are important for several reasons; including trade regulation, marketing strategy and efficient use of labor and resources [2,3].

Maturity can only be determined by a combined assessment of several parameters of nutritional quality.

Typical test parameters are dry matter content (DM) [4], titratable total acidity (A) [5], water soluble solids content (SSC)—which provides information on the sugar content of the fruit [5–7]—and, for some fruit species, total anthocyanin content (TA) [5]. The quantification of these properties is a good approximation to describe the ripeness, but since these parameters are related to each other at some level, different derived values have been developed, such as the sugar: acid ratio (SSC/A) (maturity index) and the Thiault index (TI) [8,9].

A recent study reported a system to solve the matrix dependence of spectral imaging. It presented an intelligent, all-in-one spectral imaging (ASI) laboratory system that enables standardized, automated data acquisition and real-time deployment of spectral

models. The ASI system provides a controlled, standardized illumination environment, a built-in computing system, embedded software for automated image acquisition and the deployment of models that predict the spatial distribution of sample properties in real time [10].

A summary study presented the research results of the determination of various parameters of visible/near infrared (Vis/NIR) spectroscopy of fruit samples; namely, apples, oranges, grapes, mangoes, bananas, peaches, nectarines, apricots, cantaloupes, pomegranates, dates, persimmons, pears, kiwis, strawberries and cherries, among others. The properties studied were: water soluble solids; titratable acidity; total phenols; starch iodine index; extractable anthocyanins; citric acid; potential anthocyanins; ascorbic acid; total acidity; firmness; and organic acids. There are few test results for cherries and sour cherries [11,12].

A new portable instrument based on visible and near-infrared (Vis/NIR) spectroscopy developed by the University of Bologna was used to monitor the ripening process of cherry fruits. The Cherry-Meter is used to measure the Index of Absorbance Difference (IAD), an indicator capable of monitoring the ripening process of cherry fruits. The reliability of IAD as a ripening index is confirmed by the fact that it correlates with some quality characteristics such as color (intensity of skin color), fruit anthocyanin content and soluble solids content (SSC) [13].

A non-destructive evaluation of dry matter, soluble solids and titratable acidity (A) of cherries was carried out at different temperatures in a relatively narrow measurement range of Vis/NIR. The obtained measurement results correlate well with the sensory assessment of ripeness [14].

The correlations between soluble solids content and pH of cherries at different levels of ripeness were investigated using near-infrared (NIR) hyperspectral imaging technology [6].

Using ripeness level as a quality index and total soluble solids content as a quantity index, the SeeFruits device, a low-cost, cloud-based, portable near-infrared (NIR) system for fruit quality detection, was designed [7].

Shao investigated physical damage to cherry samples using Vis/NIR techniques. He grouped samples into bruised, slightly bruised, and normal categories based on LS-SVM pattern recognition of the spectra using supervised learning. In addition to determining the degree of bruising, the maturity and the water-soluble solids were determined [15].

Information and data on the analysis of sweet cherries and cherries is summarized in Table 1.

**Table 1.** Summary of previous works dealing with the NIR analysis of cherry quality.

| Parameters | Measurement Interval, nm | Chemometric Method | Accuracy; $R^2$ | Reference |
|---|---|---|---|---|
| SSC, TA | 560; 640; 750 | n.i. | SSC/$I_{AD}$: 0.99 <br> TA/$I_{AD}$: 0.93 | [13] |
| SSC, DM, A | 729–975 | PLS | SSC *: 0.925–0.938 <br> DM *: 0.916–0.924 | [14] |
| SSC; pH | 972–1649 | PCR; PLSR | classification 96.4% | [6] |
| TSS, maturity level | "SeeFruits" | PLS; LDA; SVC; LR; LDA; PCR; LMR | classification: SVC: 0.89, Logistic-R: 0.83, LDA: 0.80 qualification: MLR: 0.77; PCR: 0.83; PLS: 0.83; SVR: 0.74 | [7] |
| bruise degree | 350–2500 | PCA, LS-SVM, SPA | LS-SVM: 93.3%; SPA: 97.3% | [15] |

n.i.: no information; * depending on temperature. LDA: linear discriminant Analysis; LR: logistic regression; LS-SVM: least square-support vector machine; MLR: Multiple linear regression; PCA: principal component analysis; PCR: principal component regression; PLS: partial least square regression; SPA: successive projection algorithm; SVC: Support vector classification; SVR: supported vector regression; TSS: total soluble solids.

The aim of this study was to develop a non-destructive, rapid measurement technique for the complex determination of sweet cherry and cherry ripeness, which is capable of both evaluation and classification, considering several characteristics. In this way, by adapting the functions to suitable hand-held and in-line equipment, it will be possible to rapidly sort fruit for quality.

## 2. Materials and Methods

### 2.1. Materials

Experiments were carried out on *Prunus avium* cv. 'Bigarreau Burlat' (32 samples) and *Prunus avium* 'Valery Chkalov' (35 samples), *Prunus cerasus* cv. 'Kántorjánosi' (31 samples) and *Prunus cerasus* cv. 'Újfehértói fürtös' (30 samples) (Szabolcs-Szatmár-Bereg regio).

The ripening period is from the end of May to the beginning of June for sweet cherries and from the end of June to the beginning and to mid-July for sour cherries.

The ripening stages were studied over two years. All varieties were followed through ripening, including immature and mature samples.

### 2.2. Methods

The following parameters were measured to characterize the ripening stage: dry matter content (DM); titratable acidity (expressed as malic acid) (A); water soluble solids (Brix°) (SSC); total anthocyanin content (TA). The maturity index was determined by calculation based on the reference values: sugar:acid ratio (SSC/A).

#### 2.2.1. Reference Methods

#### Dry Matter Content (DM)

The samples were dried under gentle conditions (70 °C) in an air-conditioned airing cupboard (Memmert, Schwabach, Germany) to constant weight. The mass was measured with analytical precision [16].

#### Titratable Acidity (A)

Total acidity was determined by potentiometric titration. Total acidity (0.1 mol/L NaOH, pH: 8.1) was calculated as mg/g or %, $w/w$ (fresh weight) malic acid [17].

Three parallel measurements were taken for each sample.

#### Water Soluble Solids—Brix° (SSC)

Brix° was determined using a refractometer calibrated for sucrose content (Pocket PAL-1, ATAGO, Tokyo, Japan). This operates in the range 0.0–53.0 Brix° with an accuracy of 0.2°. The pulped sample was centrifuged at 6000 rpm for 20 min (Micro 22R Hettich, Germany). Further analyses were performed from the supernatant, a few drops of which were transferred to the prismatic surface of the refractometer. The Brix° value of the sample was read to one decimal place [18].

Three readings per sample were taken, the measuring surface being cleaned with distilled water between each reading.

#### Anthocyanin Content (TA)

Total anthocyanin content was determined by pH differential method [19–21].

TA was determined from supernatants obtained during sample preparation as described in the SSC measurement. From the sample, 50 mL of centrifuged juice was diluted to 250 mL volume with two different buffers (pH1 and pH4.5).

The pH value of the pH1 buffer ($2.5 \times 10^{-2}$ mol/L KCl solution) solution was adjusted to pH = 1 with HCl solution.

The pH of the pH4.5 buffer solution (0.4 mol/L Na-acetate solution) was adjusted to pH = 4.5 with HCl solution.

The absorbance of the samples prepared with the two buffers was measured at wavelengths λ = 520 nm and 700 nm. Interfering components were detected at 700 nm. The absorbance of solutions were measured within twenty to fifty minutes.

The total anthocyanin content is given in cyanidin-3-glucoside equivalent, mg/L, calculated according to the following relationship:

$$\text{Abs} = (\text{Abs}_{520} - \text{Abs}_{700})_{\text{pH1}} - (\text{Abs}_{520} \cdot \text{Abs}_{700})_{\text{pH4,5}} \tag{1}$$

$$\text{TA} = \text{Abs} \cdot \text{DF} \cdot 449.2/\varepsilon \cdot l, \tag{2}$$

where:

TA = total anthocyanin content in mg/L units;

Abs520 and Abs700 are the absorbance of the same sample at two different wavelengths;

DF = dilution factor—in this case this is 5;

449.2 g/mol = molar mass of cyanidin-3-glucoside;

$\varepsilon$ = 26,900 L/(mol·cm) is the molar absorption coefficient of the solution at 520 nm;

$l$ = 1 cm is the optical path length.

Spectrophotometric measurements were performed using a Thermo Electronic UV-Vis 2.02 spectrophotometer (Thermo Fisher Scientific, Waltham, MA, USA) and the instrument's Vision Pro software. Three parallel measurements per sample were performed.

Sugar:Acid Ratio (SSC/A)—Maturity Index

The sugar:acid ratio was calculated as the ratio of the water-soluble solids content expressed in Brix° to the titratable acidity calculated in %, *w/w* [22].

### 2.2.2. FT-NIR Measurements
Sample Preparation

When designing the sample preparation, it was considered that the procedure should not require special or complicated sample preparation steps. The aim was to allow easy measurement of the samples so that quality control could be carried out in the field.

Measurements

The FT-NIR spectra were recorded, and the data processed using a Bruker MPA FT-NIR instrument (BRUKER, Ettlingen, Germany). The instrument has a scanning rate of 10 kHz. For diffuse reflectance measurements a PbS detector was used; the optical unit of the instrument is the high stability ROCKSOLID$^{\text{TM}}$ interferometer.

The diffuse reflectance spectra were recorded with a resolution of 16 cm$^{-1}$, averaging 32 sub-spectra to obtain the final spectral image.

A rotating quartz sample holder with a diameter of 85 mm was used to provide the largest possible surface area. For the rotated spectrum, the sample interacted with infrared photons not only on the surface of the 20 mm diameter detector, but in a 20 mm wide circular band.

Evaluation of FT-NIR Spectra

Spectral data were evaluated using OPUS 7.2 (Bruker, Ettlingen, Germany) and Unscrambler 10.4 (CAMO, Oslo, Norway) software.

### 2.2.3. Chemometric Methods
Principal Component Analysis—PCA

This data reduction method was used primarily for the determination of spectral outliers in NIR.

PCA is a linear unsupervised pattern recognition technique [23].

Partial Least Squares Method—PLS Regression

The PLS method is a multivariate chemometric method that is well suited for cases where there are more variables in the dataset than in the sample (reference data—as independent variables—are combined with a spectrum of nearly 2000 data points as dependent variables). PLS was applied to find the relationship between the independent variable X (reference data) and the dependent variable Y (spectral data) [24].

Several parameters were used to qualify the PLS function (Table 2).

**Table 2.** Characteristic qualifying parameters of PLS regression.

| Parameters | Calibration | Validation | Aim |
|---|---|---|---|
| | Notation | | |
| Square of the determination coefficient | $R^2$ | $Q^2$ | To be as close as possible to 1 |
| Mean squared error | RMSEC | RMSECV; RMSEP | As small as possible |
| PLS principal component | 3–10 | 3–10 | Below 3 the function is under-fitted, above 10 it is over-fitted |
| RPD—Ratio of Performance to Deviation | $(1-R^2)^{-0.5}$ | $(1-Q^2)^{-0.5}$ | if >3, the function is suitable for quantitative evaluation |
| bias | | <0.1 RMSECV; <0.1 RMSEP | To be at least one order of magnitude smaller than the average error |

Among the parameters, the mean squared error (RMSECV for cross-validation; RMSEP for test-validation) was calculated according to the following relation:

$$\text{RMSECV (RMSEP)} = \sqrt{\frac{1}{N}\sum_{i=1}^{N}\left(y_i^m - y_i^b\right)^2}, \qquad (3)$$

where:

RMSECV: root mean square error of cross-validation (the unit of measurement is the same as that of the estimated parameter);

RMSEP: root mean square error of test validation (unit of measurement equal to the estimated parameter);

$y_i^m$: measured (reference) value of the i-th component;

$y_i^b$: estimated value of the i-th component;

$N$ = number of samples tested.

The maximum number of PLS principal components was set at ten to avoid under- or over-fitting. Before setting up the PLS function, various data preprocessing and spectrum transformation procedures were set up. Considering that the surface of the fruits under investigation were covered with a thin layer of wax, it was advisable to use the 1st derivation (1st) and its combination steps in addition to the simple data treatments (standard normal variate—SNV, multiplicative scatter correction—MSC). In the combination steps, the derivation was preceded by a normalization (1st + SNV) or a multiplicative scatter correction (1st + MSC) or a correction for the slope of the spectrum (1st + SLS). The various data pretreatment procedures can be found in several publications; therefore, these are not described in detail in this article [25].

To check the resulting PLS function, cross-validation is normally used. If the sample size allows, it is worth performing a test validation using an independent dataset. The two methods will naturally yield different results, but both are qualitatively correct [26].

Linear Discriminant Analysis—LDA

Linear discriminant analysis (LDA) is a particularly popular supervised classification algorithm because it is both a classifier and a dimensionality reduction technique. It can be used to identify which features of a spectrum are most significant. LDA can also be used to classify unknown samples. For the linear discriminant analysis, we used the original spectra, and no data pre-processing was performed [27].

Generalized discriminant analysis model (GDA) is used to classify unknown samples into groups based on spectra previously established using the LDA procedure.

Quadratic Discriminant Analysis—QDA

QDA is a variant of LDA in which an individual covariance matrix is estimated for every class of observations. QDA is particularly useful if there is prior knowledge that individual classes exhibit distinct covariances, and allows for non-linear separation of data [28,29].

Discriminant Analysis Based on Mahalanobis Distance

This function performs a discriminant analysis based on the squared generalized Mahalanobis distance (D2) of the observations to the center of the groups [30].

Support Vector Machine Classification (SVM)

SVM is a linear model for classification and regression problems. It can solve linear and non-linear problems and works well for many practical problems. The idea of SVM is simple: the algorithm creates a line or a hyperplane which separates the data into classes [31].

## 3. Results

### 3.1. Reference Results

Data for the ranges of the parameters evaluated are summarized in Table 3.

**Table 3.** Concentration ranges for sweet cherries and sour cherries and reference data for ripe fruit.

| Sweet Cherry | | | |
|---|---|---|---|
| **Parameters** | **Concentration Range** | **Reference Data for Ripe Fruit** | **Reference** |
| DM; % $w/w$ | 14.70–36.01 | 20.0 | |
| A; % $w/w$ | 0.39–1.31 | 0.24 | [32–35] |
| SSC; g/100 mL | 8.7–22.4 | ≥14.8 | |
| TA; % $w/w$ | 0–158.8 | 81.2 | [36] |
| SSC/A | 10.80–36.14 | 25.4–28.7 | [37] |
| **Sour Cherry** | | | |
| **Parameters** | **Concentration Range** | **Reference Data for Ripe Fruit** | **Reference** |
| DM; % $w/w$ | 16.43–32.58 | 26.0 | |
| A; % $w/w$ | 1.34–3.04 | 1.05 | [38] |
| SSC; g/100 mL | 9.25–17.85 | ≥17.6 | |
| TA; % $w/w$ | 0–164.1 | >90 | [39,40] |
| SSC/A | 3.74–12.41 | 8.3–18.6 | [38] |

In all cases the measurement results refer to fresh fruit.

### 3.2. NIR Spectra Evaluation

Figure 1 shows the raw spectra of the two varieties of cherries and sour cherries and their first and second derivatives.

Comparing the spectra and derivatives of two different varieties of sweet and sour cherries, no differences are observed.

The characteristic vibrational ranges of organic acids (titratable acidity), water soluble solids and total anthocyanin content of the samples are summarized in Table 4 [41,42].

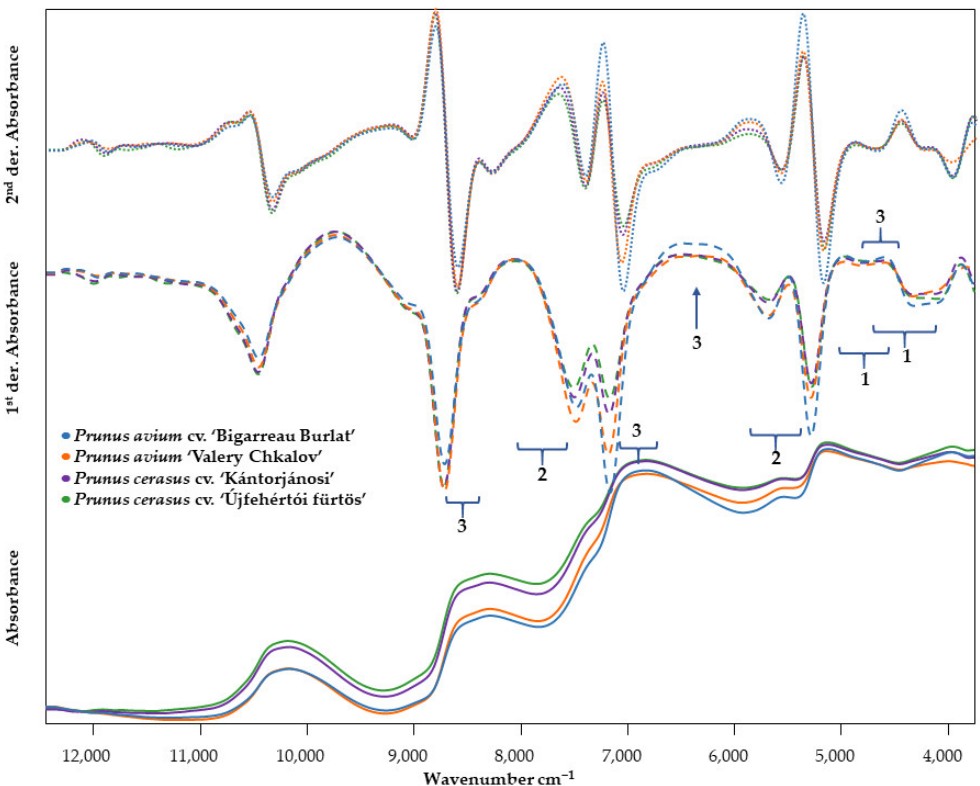

**Figure 1.** Raw, first and second derivative spectra of the sweet cherry and sour cherry varieties. (1: Soluble Solid Content, 2: Titratable Acidity, 3: Total Anthocyanin).

**Table 4.** Typical vibration ranges of tested components.

| Mark | Parameter | Wavenumber (cm$^{-1}$) |
|---|---|---|
| 1 | Soluble Solids Content (Brix°) | 4760; 4400; 4290–4250 |
| 2 | Titratable Acidity | 8100–7500; 5170–5100; 4830–4650 |
| 3 | Total Anthocyanin | 8750–8600; 7110–6900; 6850; 6370; 4750–4650; 4390–4370 |

### 3.3. Chemometric Evaluation

Chemometric evaluations (PCA, PLS, LDA, SVM) were performed using Unscrambler 10.4 (CAMO, Oslo, Norway) software.

#### 3.3.1. Principal Component Analysis—PCA

The PCA analysis was performed using the NIPALS algorithm. Ten principal components were tested. The resulting correlation was performed using seven-segment random cross-validation.

The ratio of calibrated and validated residual variance was 0.5, whereas the ratio of validated and calibrated residual variance was 0.75. The residual variance Increase limit 6.0%.

At the beginning of the maturity process, the sugar/acid ratio is low because the low sugar content and the high fruit acidity make the fruit taste sour. During the maturity process, the fruit acids break down, the sugar content increases, and the sugar/acid ratio reaches a higher value (Figure 2).

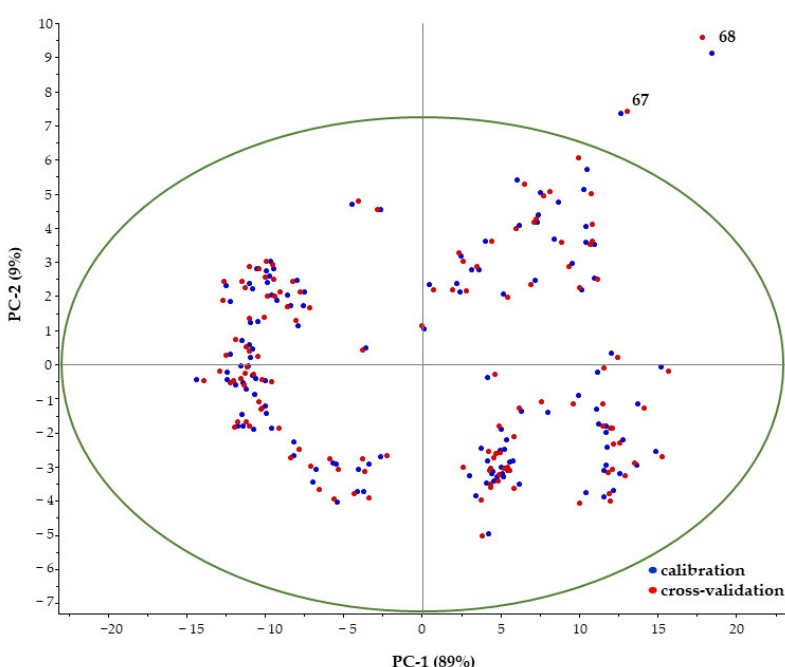

**Figure 2.** PCA analysis.

The PCA analysis shows that two samples lie outside the 95% confidence band. To verify that these are true spectral outliers, we need to examine the Leverage–F-residuals relationship (Figure 3). Samples with higher leverages have a stronger influence on the model than other samples; they may or may not be outliers, but they are influential. An influential outlier (high residual + high leverage) is the worst case; it can however easily be detected using an influence plot. Two samples (67 and 68) have high residual, but low leverage. These samples are not extreme in the model; however, they do not fit in the model well.

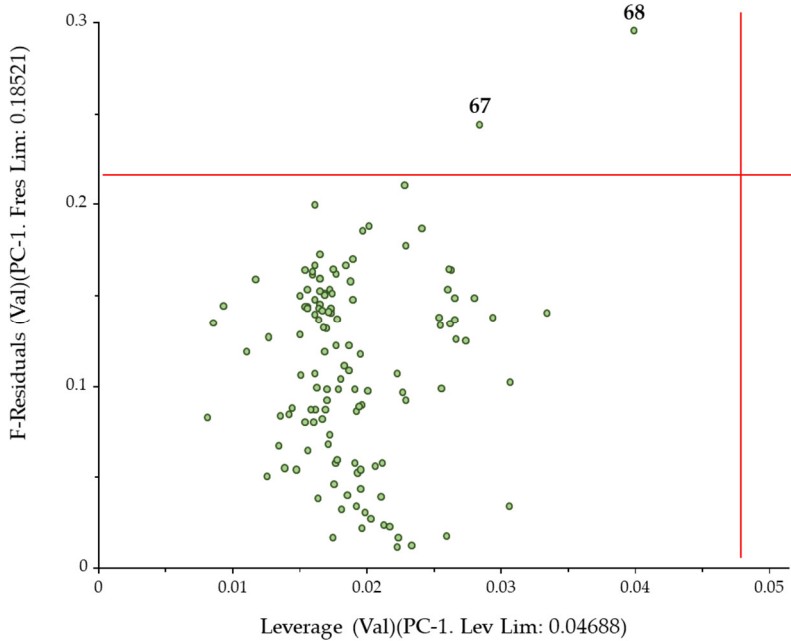

**Figure 3.** F-Residuals vs. Leverage.

### 3.3.2. Partial Least Square Regression—PLS

Two different procedures were used to check the correctness of the correlations: 7-fold cross-validation and test-validation. In the latter case the calibration: test data was split in a 2:1 ratio. The test data were selected randomly (Table 5).

**Table 5.** Statistical characteristics of PLS regression as a result of five segments cross- and test-validation.

|  | Calibration | | | Cross-Validation | | RPD | Data Preprocessing |
|---|---|---|---|---|---|---|---|
|  | $R^2$ | RMSEE | Rank | $Q^2$ | RMSECV | | |
| DM | 0.966 | 1.05 | 9 | 0.948 | 1.25 | 4.35 | 1st + SNV |
| A | 0.977 | 0.13 | 8 | 0.966 | 0.14 | 5.46 | 1st |
| SSC | 0.954 | 0.67 | 8 | 0.908 | 0.97 | 3.3 | 1st + SNV |
| TA | 0.937 | 14.0 | 7 | 0.894 | 17.5 | 3.06 | 1st + SNV |
| SSC/A | 0.959 | 1.27 | 8 | 0.925 | 1.66 | 3.92 | 1st |
|  | Calibration | | | Test Set Validation | | RPD | Data Preprocessing |
|  | $R^2$ | RMSEE | Rank | $Q^2$ | RMSEP | | |
| DM | 0.957 | 1.14 | 7 | 0.939 | 1.46 | 4.07 | 1st + SNV |
| A | 0.979 | 0.12 | 9 | 0.938 | 0.19 | 4.06 | 1st |
| SSC | 0.955 | 0.72 | 7 | 0.897 | 0.99 | 3.16 | 1st + SNV |
| TA | 0.956 | 11.6 | 7 | 0.902 | 16.9 | 3.31 | 1st + SNV |
| SSC/A | 0.959 | 1.24 | 7 | 0.939 | 1.59 | 3.82 | 1st |

In all cases the measurement results refer to fresh fruit.

RPD in Table 5, Ratio of Performance to Deviation, is the ratio of the standard error in prediction to the standard deviation of the samples, which is frequently used in NIR literature for assessing the usefulness or goodness-of-fit of calibration models. It attempts to scale the error in prediction with the standard deviation of the property (Williams 2010). Since $R^2$ ($Q^2$) and RPD are correlated, the conclusion was accepted that, if $R^2(Q^2) > 0.75$ (equal RPD > 2), then the model fits (predicts) reasonably well and if $R^2(Q^2) < 0.5$ (equal RPD < 1.4), then it does not fit as well. It is a well-established fact that if RPD > 3, the function is suitable for quantitative forecasting [43,44].

### 3.3.3. Pattern Recognition Methods

The basic aim was to separate mature and immature samples. A classification system was developed based on the literature and my own measurement results. Two maturity categories were established based on the reference values: mature and immature (Table 6).

**Table 6.** Concentration ranges for immature—mature groups.

|  | Sweet Cherry | Sour Cherry |
|---|---|---|
|  | Mature | |
| A; % *w/w* | 0.70–0.80 | 2.00–2.40 |
| SSC; g/100 mL | ≥13.0 | ≥12.4 |
| TA; % *w/w* | ≥70.0 | ≥70.0 |
| SSC/A | ≥15.0 | ≥7.1 |

In the preliminary principal component analysis, ten principal components were constructed in each case. Among the pattern recognition methods, Quadratic Discriminant Analysis (QDA) produced one of the best results of the studied parameters (Tables 7 and S1). In all cases, five-segment cross-validation was applied.

**Table 7.** Quadratic Discriminant Analysis Classification results-based raw spectra (confusion matrix).

| | Mature | Immature | Accuracy | Misclassification (Pieces) |
|---|---|---|---|---|
| **Total Titratable Acidity (A)** | | | | |
| Mature | 61 | 9 | 89.06% | 14 |
| Immature | 5 | 53 | | |
| **Soluble Sugar Content (SSC)** | | | | |
| Mature | 65 | 2 | 89.84% | 13 |
| Immature | 11 | 50 | | |
| **Total Anthocyanin (TA)** | | | | |
| Mature | 71 | 1 | 93.75% | 8 |
| Immature | 7 | 49 | | |
| **Maturity Index (SSC/A)** | | | | |
| Mature | 67 | 5 | 89.06% | 14 |
| Immature | 9 | 47 | | |

Given the strong correlation between the anthocyanin, titratable acidity and water-soluble carbohydrate content of the sample, a new category of "maturity degree" (MD) was developed.

Considering that the water-soluble solids content is crucial for maturity, we weighted the properties—80% SSC, 10% A and 10% TA were taken into account when calculating the sum.

Taking all the sub scores (A, TA, SSC) into account, it was found that if this sum was greater than 20, the sample was considered mature.

For the grouping by combined traits, we used the previously best-ranked pattern recognition analysis (QDA) (Table 8).

**Table 8.** "Maturity degree" (MD) classification model.

| **Maturity Degree (MD)** | | | | |
|---|---|---|---|---|
| | **Mature** | **Immature** | **Accuracy** | **Misclassification (Pieces)** |
| Mature | 64 | 0 | 98.44% | 2 |
| Immature | 2 | 62 | | |

Thus, an accuracy of 98.44% was achieved in sample recognition. Two samples were misclassified. One sample of sweet cherries (27th) and one sample of sour cherries (50th) should have been classified as ripe based on their MD value, but the system identified them as immature.

Examining the individual parameters, we concluded that the sweet cherry sample had an SSC just on the borderline of the category we considered ripe, while the sour cherry sample had a relatively high titratable acidity, which would have placed it in the immature category, and may have caused the misclassification.

## 4. Discussion

A non-destructive FT-NIR technique was successfully developed for the determination of titratable total acidity (Interval 0.39–3.04 %, *w/w*; RMSECV = 0.14 %, *w/w*, RMSEP = 0.19%, *w/w*), water-soluble total solids (Interval 8.7–22.4%, *w/w*; RMSECV = 0.97%, *w/w*, RMSEP = 0.99%, *w/w*), total anthocyanin content (Interval 0.1–164 g/100 mL; RMSECV = 17.5 g/100 mL, RMSEP = 16.9 g/100 mL) and the maturity index (Interval 3.7–44.2; RMSECV = 1.66, RMSEP = 1.59) of sweet cherry and sour cherry samples.

The methods were validated by seven-segment random cross-validation and test set-validation. In the latter case, the number of samples in the calibration and validation datasets was split 2:1.

On the basis of my own reference values, the typical concentration contents/interval of the mature and immature samples were determined.

Among the pattern recognition methods, different methods of linear discriminant analysis (LDA) (linear, quadratic, Mahalanobis) and linear support vector machine (SVM) were investigated.

Among different pattern recognition procedures, quadratic discriminant analysis was the one that produced favorable results for all traits.

Based on my results, a maturity scale was developed. Based on this, a 98.44% accuracy pattern recognition was achieved.

The MD $\geq 20$ value obtained in the maturity assessment (80% SSC + 10% TA + 10% A) is of course not generalizable, as it is highly species- and variety-dependent. However, as a guiding principle, it provides a broader overview of the concept of maturity than the maturity index (MI) often used.

The estimation functions allow rapid quality control and, on the other hand, the spectral database allows a rapid, even automated, identification of unripe fruits—ripe for consumption.

**Funding:** This research was funded by GINOP-2.2.1-18-2020-00025 (Development of the complex system of fruit production and processing in the Szabolcs-Szatmár-Bereg region for efficient and sustainable economic operation).

**Supplementary Materials:** The following supporting information can be downloaded at: https://www.mdpi.com/article/10.3390/pr10112423/s1, Table S1: Pattern recognition sub-scores.

**Institutional Review Board Statement:** Not applicable.

**Informed Consent Statement:** Not applicable.

**Data Availability Statement:** The data presented in this study are available on request from the author.

**Conflicts of Interest:** The author declares no conflict of interest.

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
