# Peer review of "Development of FT-NIR Technique to Determine the Ripeness of Sweet Cherries and Sour Cherries"

_processes, doi:10.3390/pr10112423_

Round 1
Reviewer 1 Report
The paper gives an estimation technique of cherry ripeness that is a very important tool in fruit harvest and quality. Further, an FIT-NIR determination is rapid, easy to use and can be achieved in field.
The paper is enough innovative and well built.
Just two observations:
Line 58. Among the studied properties vitamin C and ascorbic acid are listed but they refer to the same compound. Delete one them or replace with vitamin C or ascorbic acid.
Line 243-244. I agree with the sentence “The various data pretreatment procedures can be found in several publications”, but you must anyway cite some publications or describe the data of pretreatment procedures.
Author Response
Dear Reviewer,
Thank you very much for your review and comments on the manuscript.
In line 58, I have deleted "vitamin C"
In lines 243-244, I have provided only one reference because it thoroughly describes and explains the essence of the different data handling procedures.
This reference is Vandeginste et al.; Data Handling in Science and Technology Volume 20, Part 2, 1998, Pages 349-381

Reviewer 2 Report
The research work is good but needs certain corrections.

Author Response
Thank you for the detailed rewieve, I have answered all points in detail in the attached material. I have corrected the manuscript according to your suggestions.
The English language accuracy of the manuscript has been checked by a language proofreader
